# LncRNA GAS5 Attenuates Cardiac Electrical Remodeling Induced by Rapid Pacing via the miR-27a-3p/HOXa10 Pathway

**DOI:** 10.3390/ijms241512093

**Published:** 2023-07-28

**Authors:** Siqi Xi, Hao Wang, Jindong Chen, Tian Gan, Liang Zhao

**Affiliations:** Department of Cardiology, Shanghai Chest Hospital, School of Medicine, Shanghai JiaoTong University, Shanghai 200003, China; xisiqi@sjtu.edu.cn (S.X.); wh2071@shchest.org (H.W.); chenjindong@shchest.org (J.C.); ganruoning@sjtu.deu.cn (T.G.)

**Keywords:** lncRNA GAS5, electrical remodeling, ion channel, rapid pacing

## Abstract

Previous studies indicated long non-coding RNAs (lncRNAs) participated in the pathogenesis of atrial fibrillation (AF). However, little is known about the role of lncRNAs in AF-induced electrical remodeling. This study aimed to investigate the regulatory effect of lncRNA GAS5 (GAS5) on the electrical remodeling of neonatal rat cardiomyocytes (NRCMs) induced by rapid pacing (RP). RNA microarray analysis yielded reduced GAS5 level in NRCMs after RP. RT-qPCR, western blot, and immunofluorescence yielded downregulated levels of Nav1.5, Kv4.2, and Cav1.2 after RP, and whole-cell patch-clamp yielded decreased sodium, potassium, and calcium current. Overexpression of GAS5 attenuated electrical remodeling. Bioinformatics tool prediction analysis and dual luciferase reporter assay confirmed a direct negative regulatory effect for miR-27a-3p on lncRNA-GAS5 and HOXa10. Further analysis demonstrated that either miR-27a-3p overexpression or the knockdown of HOXa10 further downregulated Nav1.5, Kv4.2, and Cav1.2 expression. GAS5 overexpression antagonized such effects in Nav1.5 and Kv4.2 but not in Cav1.2. These results indicate that, in RP-treated NRCMs, GAS5 could restore Nav1.5 and Kv4.2 expression via the miR-27a-3p/HOXa10 pathway. However, the mechanism of GAS5 restoring Cav1.2 level remains unclear. Our study suggested that GAS5 regulated cardiac ion channels via the GAS5/miR-27a-3p/HOXa10 pathway and might be a potential therapeutic target for AF.

## 1. Introduction

Atrial fibrillation (AF) is the most common sustained cardiac arrhythmia in adults and is one major cause of stroke and heart failure, causing increased morbidity and mortality [1]. Owing to extended longevity in the general population and the improved detection of AF, the estimated AF risk was one in three individuals of ancestry at an index age of 55 years by 2020 [2]. The temporal pattern of AF is crucial for the evaluation of AF severity; generally, AF progression is characterized by increased onset frequency and duration, along with deteriorated clinical symptoms [2].

One major mechanism of AF persistence is the electrical remodeling of cardiomyocytes [3,4,5], which is largely thought to be induced by rapid atrial rates. Electrical remodeling encompasses changes in the properties of atrial ion channels, subsequently affecting myocardial activation and conduction [6,7]. This process is characterized by an abbreviation of action potential duration (APD) and shortened refractoriness [5,8], which has been observed soon after the application of rapid stimulation in in vitro studies [3]. However, due to the complexity of the AF mechanism, our understanding of the pathogenesis of electrical remodeling during AF remains limited.

Long non-coding RNAs (lncRNA), a large class of more than 200 non (protein)-coding RNAs [9], could regulate gene expressions by interacting with DNAs, mRNAs, miRNAs, or proteins [10,11] and play important roles in cell-cycle control, cell stability, cell differentiation, transcription, and translation [12]. Previous studies have shown that lncRNAs participate in cardiac development as crucial mediators [13], and the regulatory role of lncRNAs has also been observed in cardiac hypertrophy [14], myocardial infarction, and coronary heart disease [15]. Yet few studies have focused on the role of lncRNAs in the pathogenesis of AF.

LncRNA growth arrest-specific 5 (lncRNA GAS5, GAS5), first discovered in 1988 [4], is widely considered to be a suppressor in the proliferation, apoptosis, migration, and invasion of tumor cells via multiple mechanisms such as methylation, autophagy, and sponging target molecules [16,17,18,19]. Recently, a growing body of evidence has indicated that GAS5 is also involved in cardiovascular pathologies such as cardiac hypertrophy [20], myocardial infarction [21], and myocardial apoptosis [22]. Furthermore, GAS5 has been reported to be a novel biomarker for AF [23], suggesting a potential link between GAS5 and AF. Yet it remains unclear if GAS5 plays a regulatory role in the pathogenesis of AF.

In the present study, we hypothesized that GAS5 regulates electrical remodeling induced by rapid pacing (RP) via the miR-27a-3p/HOXa10 pathway and used neonatal rat cardiac myocytes (NRCMs) to verify RP-induced cardiac electrical remodeling and explore the underlying mechanism.

## 2. Results

### 2.1. The Expression of GAS5 Decreased in NRCMs after RP

The systematic experimental and theoretical approaches are illustrated in Figure 1. By performing differential analysis using data from GSE10598, we identified four lncRNAs differentially expressed in NRCMs after RP, including pvt1, Map2k3os, Dancr, and GAS5. Then, the downregulation of GAS5 in NRCMs was detected via RT-qPCR analysis after RP for 6 h (Figure 2A), suggesting potential associations between GAS5 level and RP.

### 2.2. The Impact of RP on Cav1.2, Nav1.5, and Kv4.2 Expression

The action potential of cardiomyocytes is initiated by Nav1.5, which conducts Na^+^ inward current and generates the upstroke of action potential [24]; Kv4.2 conducts I_to_, which is responsible for rapid repolarization [25], and Cav1.2 conducts I_CaL_, one major current of slow repolarization [26]. Therefore, we focused on the expression of these ion channels. RT-qPCR was performed to evaluate the impact of RP on Cav1.2, Nav1.5, and Kv4.2 expression in cultured primary NRCMs. (Figure 2B) Compared with the controls, RP induced the downregulation of Cav1.2, Nav1.5, and Kv4.2, as yielded by RT-qPCR (Figure 2C). Western blot analysis further verified the downregulation of ion channels after RP (Figure 2D). These results indicated that the RP treatment could induce electrical remodeling in primary NRCMs.

### 2.3. Overexpression of GAS5 Mitigated RP-Induced Ion Channel Remodeling in NRCMs

As shown in Figure 3A, the overexpression of GAS5 after transfection with pcDNA-GAS5 was verified via RT-qPCR analysis. RP induced the downregulation of Cav1.2, Nav1.5, and Kv4.2 in NRCMs, which was mitigated by the overexpression of GAS5, verified via RT-qPCR (Figure 3B), western blot analysis (Figure 3C), and immunofluorescence assay (Figure 3D).

In order to verify the regulatory role of GAS5 in ion channel remodeling, we further performed GAS5 knockdown by using siRNA and examined its impact on calcium, potassium, and sodium currents via whole-cell patch-clamp. The most efficient siRNA (siGAS5#2) was detected via RT-qPCR (Figure 3E). As yielded via whole-cell patch-clamp, RP induced a remarkable decrease in peak calcium, sodium, and potassium currents, while the overexpression of GAS5 partially restored these currents to various degrees. The knockdown of GAS5 induced a further decrease in peak I_CaL_, I_Na_, and I_to_ in NRCMs treated with RP. (Figure 3F–K)

### 2.4. GAS5 Directly Targets miR-27a-3p

It is well known that lncRNAs exert their functions in the cytoplasm via the competing endogenous RNA (ceRNA) mechanism, mostly by binding miRNA [27]. Since GAS5 is distributed in the cytoplasm [23], we performed miRNA sequencing to identify the target miRNA of GAS5 by screening differentially expressed miRNAs between the control group and GAS5 overexpression group (Figure 4A). Then, the top four downregulated miRNAs were validated via RT-qPCR (Figure 4B), among which only miR-27a-3p was upregulated after RP. (Figure 4C). Therefore, miR-27a-3p was selected as the downstream target, and the expression of miR-27a-3p in NRCMs after transfection is illustrated in Figure 4D. Then, the starBase (http://starbase.sysu.edu.cn/ (accessed on 11 January 2022)) and TargetScan (http://www.targetscan.org/ (accessed on 11 January 2022)) tools were used to predict the binding site to verify miR-27a-3p as the direct target of GAS5.

Then, 293T cells were transfected with miR-NC or miR-27a-3p mimics followed by the introduction of GAS5 wild-type or GAS5 mutant dual luciferase reporter plasmid. Data from the dual luciferase reporter assay revealed that miR-27a-3p mimics inhibited the luciferase activities of GAS5 wild-type but not GAS5 mutant (Figure 4E), suggesting that GAS5 targeted miR-27a-3p. These results suggested that GAS5 could directly regulate miR-27a-3p expression.

### 2.5. GAS5 Mitigated RP-Induced Electrical Remodeling in NRCMs via Targeting miR-27a-3p

We then determined whether GAS5 regulated the RP-induced downregulation of ion channels by targeting miR-27a-3p. As yielded by RT-qPCR (Figure 5A), western blot (Figure 5B), and immunofluorescence assay (Figure 5C), miR-27a-3p mimics induced the downregulation of Nav1.5 and Kv4.2 in NRCMs under RP, which was antagonized by GAS5 overexpression. However, miR-27a-3p seemingly did not affect the expression of Cav1.2, suggesting regulatory mechanisms of Cav1.2 other than miR-27a-3p. These results suggested that GAS5 could mitigate the downregulation of Nav1.5 and Kv4.2 induced by RP via targeting miR-27a-3p.

### 2.6. GAS5/miR-27a-3p Axis Regulated Electrical Remodeling through Hoxa10

TargetScan database predictive analysis revealed the presence of a miR-27a-3p binding site in the 3′UTR region of HOXa10. (Figure 6A,B) As yielded by RT-qPCR, the expression of HOXa10 mRNA decreased in NRCMs after RP for 6 h, and transfection with miR-27a-3p mimics further downregulated after RP, suggesting that miR-27a-3p could inhibit HOXa10 expression (Figure 6C). Then, we assessed the interference efficiency of siRNA HOXa10#1, siRNA HOXa10#2, and siRNA HOXa10#3 in cardiomyocytes using RT-qPCR and western blot and selected siRNA HOXa10#2 for the subsequent experiments based on its highest efficiency of interference (Figure 6D). Luciferase experimental results showed that luciferase activity decreased after the co-transformation of HOXa10 with the wild-type miR-27a-3p mimic but did not significantly decrease after co-transfection with the mutant miR-27a-3p mimic (Figure 6E). In NRCMs treated with RP, GAS5 overexpression could upregulate the expression of Nav1.5 and Kv4.2, whereas knockdown of HOXa10 downregulated the expression of Nav1.5 and Kv4.2, as yielded by western blot (Figure 6F), RT-qPCR (Figure 6G), and immunofluorescence assay (Figure 6H). Notably, the knockdown of HOXa10 plus GAS5 overexpression induced the further upregulation of Kv4.2 compared with the GAS5 overexpression group, suggesting additional regulatory mechanisms of Kv4.2 (other than the GAS5/miR-27a-3p/HOXa10 pathway).

## 3. Discussion

In the present study, as shown in Figure 7, we observed that RP-induced electrical remodeling in NRCMs manifested as a down-regulated expression of ion channels, including Cav1.2, Nav1.5, and Kv4.2, and concurrent downregulated expression of GAS5. Moreover, we provided evidence suggesting that the overexpression of GAS5 ameliorated RP-induced electrical remodeling of atrial cardiomyocytes, possibly via the GAS5/miR-27a-3p/HOXa10 pathway, suggesting the possibility of a potential therapeutic target for AF, which warrants further studies for verification.

### 3.1. Electrical Remodeling during AF

Electrical remodeling during AF, which is the primary manifestation of atrial remodeling, mainly included the altered expression of ion channels and cellular electrophysiological characteristics. Atrial RP could increase the susceptibility to AF via electrical remodeling [5,6], such as abnormal calcium handling, which increases ectopic activity and shortens atrial APD and ERP [28,29], and reduced gap junctions, which promoted atrial conduction heterogeneity [30]. In RP canine models, a reduced density of I_to_ and I_CaL_ channels was observed without changes in their function, resulting in a decreased atrial effective refractory period and APD [8,31]. In the present study, the expressions of Cav1.2, Nav1.5, and Kv4.2 were downregulated in NRCMs after RP, consistent with the pathological process of atrial remodeling during AF.

### 3.2. The Role of GAS5 in Atrial Electrical Remodeling

Previous studies have shown that GAS5 plays a regulatory role in the pathological process of a series of cardiovascular diseases such as hypertrophic cardiomyopathy [20] and myocardial infarction [21], yet its role in AF remains unclear. In the present study, the bioinformatic analysis suggested that GAS5 expression was associated with the electrical remodeling of cardiomyocytes, and our in vitro study found that the expression of GAS5 was downregulated after RP. To the best of our knowledge, this is the first time that the protective role of GAS5 has been identified in a RP model, and the overexpression of GAS5 could partly reverse RP-induced downregulation of Cav1.2, Nav1.5, and Kv4.2 on both transcriptional and translational levels. Further studies are warranted to verify these findings.

### 3.3. Electrical Remodeling via GAS5/miR-27a-3p/HOXa10 Pathway

Recently, the prominent role of the miR-27a-3p/HOXa10 pathway in the regulation of cardiovascular diseases has been observed in multiple studies. It has been reported that miR-27a-3p is closely related to ventricular formation, obesity, and cardiac effects in mice [32,33]. Previous studies have proved that a downregulated expression of miR-27a-3p can reduce cardiomyocyte injury induced by hypoxia/reoxygenation and lipopolysaccharide [34,35]. Moreover, miR-27a-3p regulates Ang II-induced cardiomyocyte hypertrophy and electrical remodeling through HOXa10, resulting in a decreased expression of Kv4.3 [36].

HOXa is involved in the regulation of embryonic development, cell differentiation, cell cycle, apoptosis, and other biological processes [37], and HOXa10 is closely associated with the occurrence and development of cardiovascular diseases [38,39]. By analyzing the expression levels of ion channels and GAS5/miR-27a-3p/HOXa10, our study, for the first time, demonstrates that the overexpression of GAS5 mitigates the downregulation of Nav1.5, Kv4.2, and Cav1.2 induced by RP. Nav1.5 and Kv4.2 was regulated via the miR-27a-3p/HOXa10 pathway, suggesting the cardioprotective potential of GAS5 against rapid cardiac electrical activity, probably including AF. However, GAS5 seemingly affected the expression of Cav1.2 via mechanisms other than miR-27a-3p, and the regulatory effect of GAS5 on Kv4.2 might involve additional mechanisms beyond miR-27a-3p, which warrants further study.

Currently, the role of lncRNA in the pathogenesis of AF remains largely unclear, and to the best of our knowledge, for the first time, our study reports the regulatory role of GAS5 in the electrical remodeling of cardiomyocytes. Our research consisted of an exploratory study conducted through in vitro experiments by using neonatal rat cardiomyocytes to investigate the impact of GAS5 and its downstream molecules on the expression of cardiac ion channels, and it yielded preliminary positive results, suggesting the possibility of extrapolation to other species. However, there are some limitations that must be taken into consideration for the interpretation of the results: (1) Considerable studies are warranted to fill the gap between small rodents and humans, such as animal experiments using canines. (2) Although the findings of our study suggested a potential association between GAS5 and myocardial electrical remodeling, extensive studies are needed to verify its practical significance as a therapeutic target and biomarker for AF, given the fact that various targeted therapeutic approaches have been introduced but are severely limited by side effects or complications [40]. (3) The mechanism of AF is complex and involves, electrical remodeling, structural remodeling, and the autonomous nerve system [31]. When concomitant diseases such as heart failure are present, the pathogenesis of AF tends to involve multiple mechanisms and such diversity could be enhanced during the progression of AF. Our study observed a positive association between GAS5 and AF, possibly via the miR-27a-3p/HOXa10 pathway. Yet its regulatory effect on electrical remodeling and consequently on the pathogenesis of AF still warrants considerable studies under different concomitant diseases and at different stages of AF (such as paroxysmal and chronic AF). In future studies, we will continue our investigation and prudently verify the regulatory effect of GAS5 on cardiac electrical remodeling in larger AF mammal models and on the pathogenesis of AF.

## 4. Materials and Methods

### 4.1. Isolation of Neonatal Rat Cardiac Myocytes (NRCMs)

NRCMs were prepared as previously described [14]. Briefly, the hearts were removed from two-day-old Sprague Dawley rats, the tissue was finely minced followed by sequential digestion with 1 mg/mL collagenase I (Worthington, Freehold, NJ, USA) and 0.125% trypsin (Sigma, St. Louis, MI, USA). The cardiomyocytes were separated from fibroblasts by differential plating and then cultured in gelatin-coated tissue culture plates in media containing Dulbecco’s modified Eagle’s medium, 10% fetal bovine serum, 100 μM bromodeoxyuridine 1% penicillin-streptomycin, and 1% L-glutamine.

### 4.2. Stimulation of Cells in Culture

Gelatin/fibronectin-coated 35 mm culture dishes were seeded with NRCMs at a density of approximately 1 × 10^6^ cells per ml so that the dishes were ≥90% confluent at 96 h. At this confluency, microscopic visualization of a coverslip demonstrated regions of spontaneous beating (50% of the total area) at a rate of approximately 60–100 beats per minute (bpm). On day 5, cultured NRCMs were transferred to serum-free medium and subjected to rapid electrical stimulation (RES) at 3.0 Hz for 360 min. RES was applied by field stimulation via two graphite electrodes (2.5 mm in diameter) oriented in parallel at the outer edges with an interpolar distance of 2 cm. A biphasic waveform was delivered by a custom-built stimulator, and the stimulation intensity was adjusted to a level of 1.5× threshold for synchronous contraction [41].

### 4.3. RT-qPCR

Total RNAs were extracted by Trizol (Invitrogen, Carlsbad, CA, USA) and reverse-transcribed into complementary DNA (cDNA) using the PrimeScript™ RT Master mix (Takara, Otsu, Japan) and PrimeScript RT reagent Kit (Takara, Otsu, Japan), according to the manufacturer’s protocol. Quantitative real-time PCR was performed on an ABI 7500 PCR instrument (Applied Biosystems, San Francisco, CA, USA) with a SYBR green Kit (Takara, Otsu, Japan). GAPDH (for mRNA) and U6 (for miRNA) were used as the reference gene. The 2^−∆∆Ct^ method was used to calculate the relative RNA levels [42]. The primer sequences are listed in Table 1.

### 4.4. Western Blot Analysis

The total proteins of the cells were lysed in a RIPA buffer containing 1% protease inhibitors (Beyotime, Shanghai, China) and collected. After SDS-PAGE, the proteins were transferred onto PVDF membranes. The primary antibodies were anti-Cav1.2 (Alomone, Jerusalem, Israel), anti-Nav1.5 (Alomone, Jerusalem, Israel), anti-Kv4.2 (Abcam, Cambridge, England), anti-HOXa10 (ABclonal, Wuhan, China), and anti-β-Actin (ABclonal, Wuhan, China). The HRP-conjugated secondary antibody HRP-labeled goat anti-rabbit IgG (Absin, Shanghai, China) was used.

### 4.5. Immunofluorescence Assay

NRCMs were washed 2 times with phosphate-buffered saline (PBS) and fixed with cold 4% paraformaldehyde for 15 min. Then, the NRCMs were blocked with a blocking solution (2% bovine serum albumin at 37 °C for 0.5 h and washed 3 times with PBS). Next, the NRCMs were incubated with Cav1.2 antibodies (Alomone, Jerusalem, Israel), Nav1.5 antibodies (Alomone, Jerusalem, Israel), and Kv4.2 antibodies (Abcam, Cambridge, England) at 4 °C for 12 h. After washing three times with PBS, the NRCMs were incubated with CY3-labeled (red fluorescence) goat anti-mouse IgG antibodies (Beyotime, Shanghai, China) at 37 °C for 1 h. The cells were kept from light, stained with DAPI (blue fluorescence) for 5 min, and washed with PBS three times [43]. Images were captured under a Zeiss LSM880 microscope with a 20 × objective (Carl Zeiss, Oberkochen, Germany).

### 4.6. Dual-Luciferase Reporter Assays

A sequence containing the potential binding site of GAS5 and the 3′UTR of HOXa10 and the corresponding mutant sequence were synthesized and cloned into the pGL3 luciferase reporter vector (Promega, Madison, WI, USA). HEK-293T cells were transfected with the plasmids described above. All luciferase activities were measured using the Dual-Luciferase Reporter Assay System (Promega, Madison, WI, USA) and normalized to Renilla luciferase activity [44].

### 4.7. Plasmid Construction and Cell Transfection

Lipofectamine 3000 (Invitrogen, USA) was used to transfect the siRNAs, miRNA mimics, and plasmids into NRCMs and HEK-293T. Full-length complementary cDNAs of GAS5 were synthesized and inserted into the expression vector pcDNA3.1 (Bioegene, Shanghai, China). NRCMs and HEK-293T for 48 h were chosen to establish stable overexpression and knockdown cell lines [45].

### 4.8. RNA Isolation, Small RNA Library Construction, and Deep Sequencing

A total of 6 samples (isolated from the NRCMs) were used for total RNA extraction using the TruSeq Small RNA Sample Prep Kits (Illumina, San Diego, CA, USA) according to the manufacturer’s protocol. The quality of the RNA samples was examined using a NanoDrop 2000 spectrophotometer (Thermo Scientific, Waltham, MA, USA) and standard denaturing agarose gel electrophoresis. Small RNA library preparation was performed using TruSeq Small RNA Sample Prep Kits (Illumina, San Diego, CA, USA). The quality-ensured RNA-seq libraries were then sequenced using Illumina Hiseq2000/2500. The identification of known miRNAs (mapped to the miRbase database) and read counting were processed using ACGT101-miR (LC Sciences, Houston, TX, USA). A modified normalization was used to correct copy numbers among different samples, and a miRNA was considered present when the normalized read count was >0 in all the samples. A heatmap was constructed using the normalized read counts of the known miRNAs in each EV sample using R (R version 4.0.3) with a heatmap via a custom written R script [46].

### 4.9. Target Gene Prediction and Pathway Enrichment Analysis

TargetScan (v5.0) [17] and miRanda (v3.3a) [18] were used to predict the target genes of miRNAs. The predicted target genes were screened according to the scoring criteria of each software. The TargetScan algorithm removes target genes whose context score percentile is less than 50, and the miRanda algorithm removes target genes whose max energy is greater than 10. The intersection of these two databases was taken as the target genes of the miRNA. Gene ontology (GO) and Kyoto Encyclopedia of Genes and Genomes (KEGG) pathway enrichment analysis of these target genes was annotated.

### 4.10. Cellular Electrophysiology Recording

Whole-cell patch-clamp was performed using the EPC-9 amplifier (Stuttgart, Germany), and data were analyzed with the Pulse-fit software interface (Version 8.31, HEKA Co., Stuttgart, Germany). The resistances of the pipettes ranged from 3 to 6 MΩ when filled with pipette solution. Series resistance (Rs) was between 4 and 10 MΩ, and compensation was applied to reduce Rs by 80–90%. Current signals were filtered at 3 kHz by an 8-pole Bessel filter, digitized at a sampling rate of 1 kHz, and stored on the computer running Pulse software, which was used for the generation of pulses [47].

Transient outward potassium current (I_to_) was measured using the whole-cell voltage-clamp mode. The external solution contained (in mM): NaCl (135), KCl (4), CaCl_2_, (1.8) MgCl_2_ (1), C_6_H_12_O_6_ (10), and HEPES (10).The pH was adjusted to 7.4 with NaOH. The pipette solution contained (in mM): KCl(145), MgCl_2_(1), EGTA(5), HEPES (10), and MgATP (5). The pH was adjusted to 7.2 with KOH. In addition, 0.005 mM Nifedipine was used to eliminate L-type calcium current. Current–voltage curve and activation were measured from a holding potential of −80 mV and were elicited with steps of 10 mV from −60 mV to +60 mV with a cycle length of 500 ms. A 100 ms pre-pulse to −40 mV from a holding potential of −80 mV was used to inactivate sodium current.

L-type calcium current (I_CaL_) was measured using the whole-cell voltage-clamp mode. The external solution contained (in mM): choline chloride (100), NaCl (35), NaH_2_PO_4_ (0.33), MgCl_2_ (1), KCl (5.4), CaCl_2_ (1.8), HEPES (10), glucose (10), BaCl_2_ (0.1), 4-aminopyridine (5), and the pH was adjusted to 7.4 with NaOH. The pipette solution contained (in mM): CsCl (120), EGTA (10), CaCl_2_ (1), MgCl_2_ (5), Na_2_ATP (5), HEPES (10), and pH was adjusted to 7.2 with CsOH. Current–voltage curve and activation were measured from a holding potential of −90 mV and were elicited with steps of 10 mV from −40 mV to +60 mV with a cycle length of 500 ms. A 300 ms pre-pulse to −40 mV from a holding potential of −90 mV was used to inactivate sodium current.

Sodium currents were measured using the whole-cell voltage-clamp mode. The external solution contained (in mM): NaCl (135), KCl (4), CaCl_2_ (1.8), MgCl_2_ (1), C_6_H_12_O_6_ (10), and HEPES (10). The pH was adjusted to 7.4 with NaOH. The pipette solution contained (in mM): NaF (5), CsF (110), CsCl (20), EGTA (10), and HEPES (10). The pH was adjusted to 7.4 with CsOH. In addition, Nifedipine (0.005 mM) and 4-aminopyride (0.5 mM) were used to eliminate I_CaL_ and I_to_. Current–voltage curve and activation were measured from a holding potential of −120 mV and were elicited with steps of 5 mV from −90 mV to +60 mV with a cycle length of 500 ms. Data analysis was performed using Clamp-fit 10.7 and Origin 9.0. All patch-clamp experiments were performed at 34–36 °C.

### 4.11. Statistical Analysis

SPSS version 21.0 (IBM Corp., Armonk, NY, USA) software was used for statistical analysis. Continuous data are presented as mean ± standard deviation (SD). Statistical analyses were performed using Student’s *t*-test and one-way analysis of variance (ANOVA), depending on the purpose and type of data.* *p*-value < 0.05 was considered significant.

## 5. Conclusions

GAS5 is involved in myocardial electrical remodeling induced by RP. Mechanistically, a GAS5/miR-27a-3p/HOXa10 signaling pathway mediates the regulatory effect of GAS5 on cardiac ion channels. Yet there are still some limitations to the present study, such as the in vitro study design and lack of data from large mammals. Therefore, the extrapolation of our findings should be prudently carried out, and further studies are warranted to verify our findings in larger mammals and explore its clinical prospect. Nevertheless, our results unveil new functions of GAS5 and provide a theoretical basis for a new possible therapeutic target for AF.

## Figures and Tables

**Figure 1 ijms-24-12093-f001:**
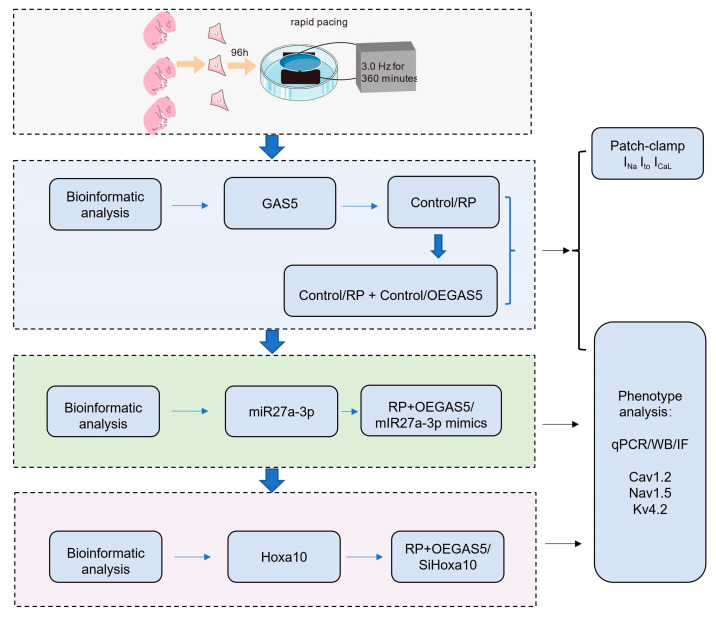
A schematic diagram showing the flow of the systematic experimental and theoretical approaches used in this study. RP, rapid pacing; OEGAS5, overexpression of GAS5; qPCR, quantitative polymerase chain reaction; WB, western blotting; IF, immunofluorescence.

**Figure 2 ijms-24-12093-f002:**
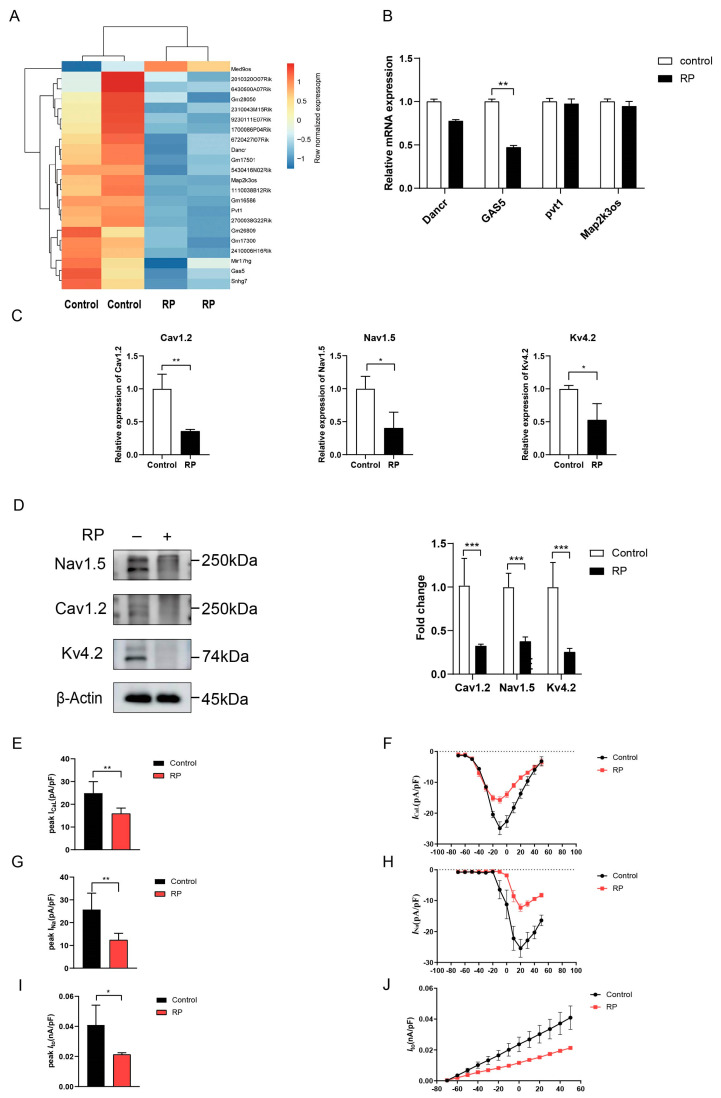
The expression of GAS5 was downregulated in cultured primary neonatal rat cardiac myocytes (NRCMs) after RP for 6 h. (**A**) Heat map of differentially expressed lncRNAs in control and RP groups. Data was retrieved from the GEO dataset (accession number GSE10598). Expression values are represented in shades of red and blue, which indicate expression above and below the median expression value across all samples. (**B**) RT–qPCR showing that GAS5 was downregulated in NRCMs among the four differentially expressed lncRNAs after RP. The expressions of Cav1.2, Nav1.5, and Kv4.2 in NRCMs were downregulated after RP, as yielded by RT-qPCR (**C**) and western blot (**D**). The peak current and current density–voltage (I–V) curve of I_CaL_(**E**,**F**), I_Na_ (**G**,**H**), and I_to_ (**I**,**J**) also decreased after RP, as yielded by whole-cell patch-clamp. Data are presented as mean ± SD. * *p* < 0.05, ** *p* < 0.01, and *** *p* < 0.001 versus control group, respectively. RP, rapid pacing; NRCMs, neonatal rat cardiac myocytes; mRNA, messenger RNA; RT–qPCR, quantitative real-time polymerase chain reaction.

**Figure 3 ijms-24-12093-f003:**
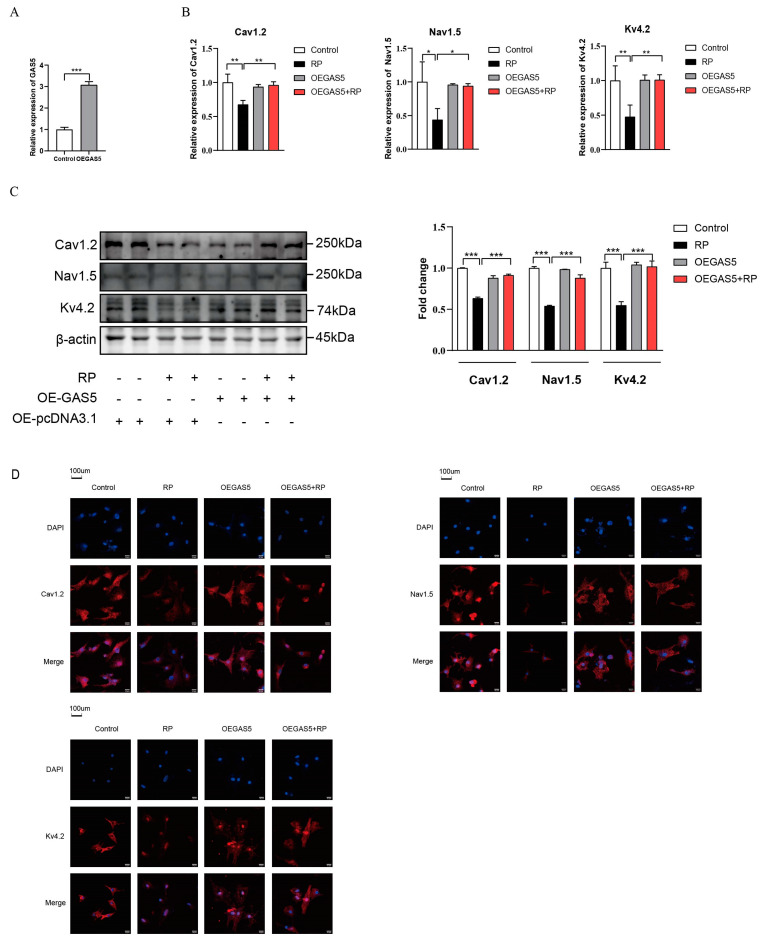
The impact of GAS5 overexpression on ion channeling remodeling induced by RP in NRCMs. (**A**) The overexpression of GAS5 was induced in NRCMs via the transfection of pcDNA-GAS5. *** *p* < 0.001 vs. control group. The overexpression of GAS5 mitigated the downregulation of Cav1.2, Nav1.5, and Kv4.2 channels in NRCMs induced by RP, as assessed by RT–qPCR (**B**), western blot (**C**), and immunofluorescence assay (**D**). (**E**) GAS5 knockdown efficiency detection. Compared with the controls, the peak current and current density–voltage (I–V) curve of I_CaL_ (**F**,**G**), I_Na_ (**H**,**I**), and I_to_ (**J**,**K**) was downregulated after RP, mitigated by GAS5 overexpression, and further downregulated by GAS5 knockdown, as yielded via whole-cell patch-clamp. Data are presented as mean ± SD. * *p* < 0.05, ** *p* < 0.01, *** *p* < 0.001. RP: rapid pacing; NRCMs: neonatal rat cardiac myocytes; mRNA: messenger RNA; RT–qPCR: quantitative real-time polymerase chain reaction; OE: overexpression.

**Figure 4 ijms-24-12093-f004:**
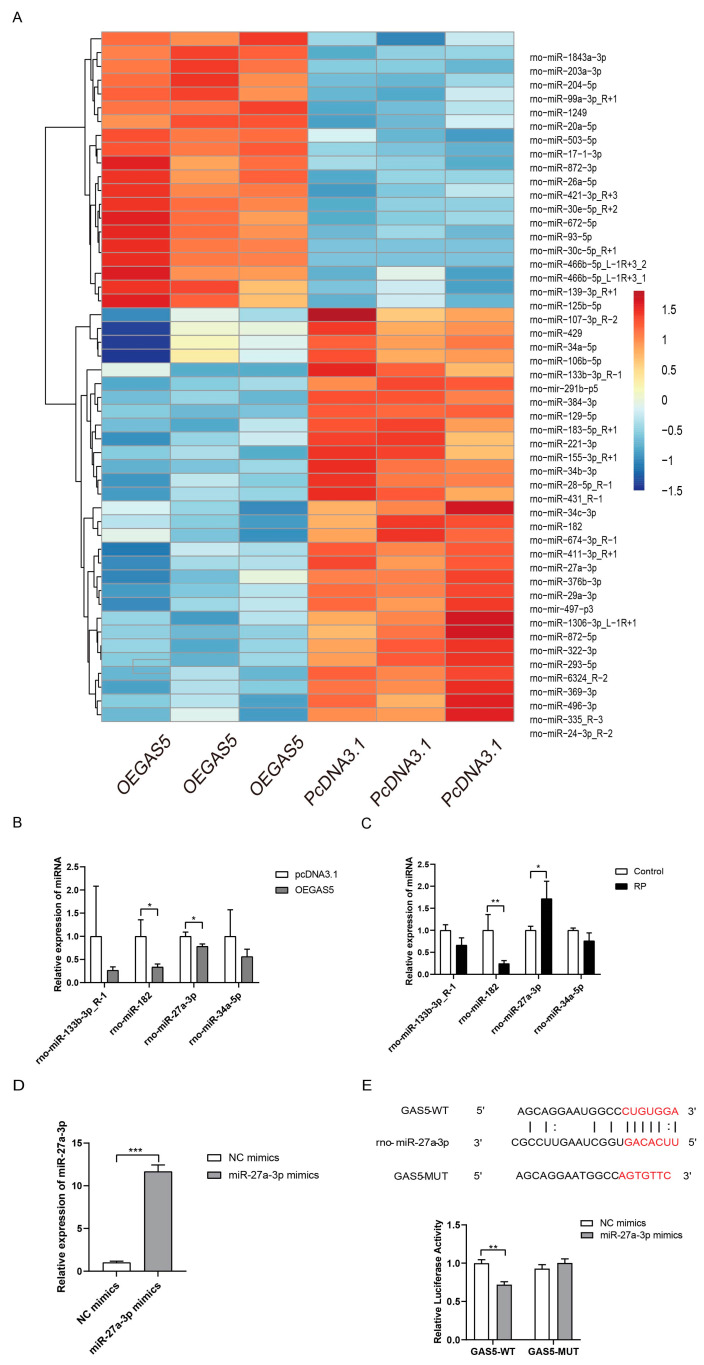
GAS5 targets miR-27a-3p directly. (**A**) A heatmap of differentially expressed miRNAs from miRNA sequencing data in NRCMs transfected with empty vector and pcDNA-GAS5. Expression values are represented in shades of red and blue, which indicate expression above and below the median expression value across all samples. (**B**) Validation of the top four significantly downregulated miRNAs in GAS5-overexpressed NRCMs. (**C**) Altered expressions of miRNAs induced by RP in NRCMs. (**D**) Increased miR-27a-3p expression in NRCMs treated with miR-27a-3p mimic. (**E**) Luciferase activity in HEK-293T cells co-transfected with GAS5 wild-type or mutant sequence and miR-27a-3p mimics. The red part indicated the binding site of GAS5wild-type and miR-27a-3p, and the GAS5 mutant sequence. Data are presented as the mean ± SD of four independent experiments. * *p* < 0.05, ** *p* < 0.01, *** *p* < 0.001 vs. NC mimic group. RP, rapid pacing; NC, negative control; OEGAS5, overexpression of GAS5.

**Figure 5 ijms-24-12093-f005:**
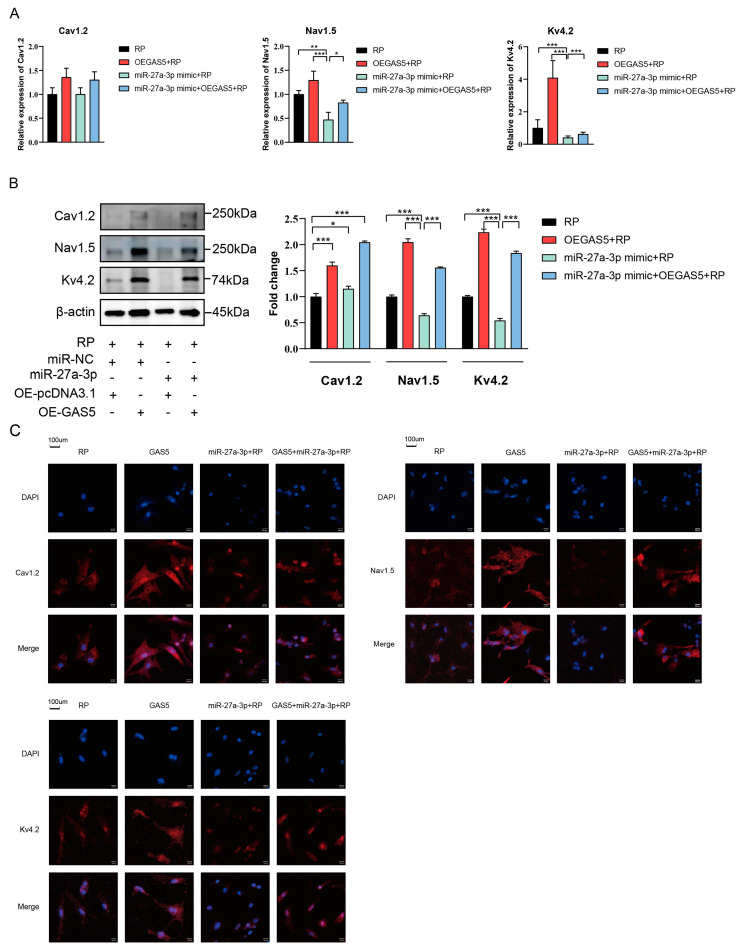
GAS5 regulated RP-induced electrical remodeling in NRCMs by targeting miR-27a-3p. RT–qPCR (**A**), western blot (**B**), and immunofluorescence assay (**C**) showing the expression levels of Cav1.2, Nav1.5, and Kv4.2 in NRCMs transfected with empty-vector, pcDNA-GAS5, miR-27a-3p mimics or co-transfected with miR-27a-3p mimics and pcDNA–GAS5 under RP. Data are presented as mean ± SD from at least three independent experiments. OEGAS5: overexpression of GAS5; OE-pcDNA3.1: overexpression of empty plasmid; NC, negative control. * *p* < 0.05, ** *p* < 0.01, *** *p* < 0.001 vs. NC mimic group.

**Figure 6 ijms-24-12093-f006:**
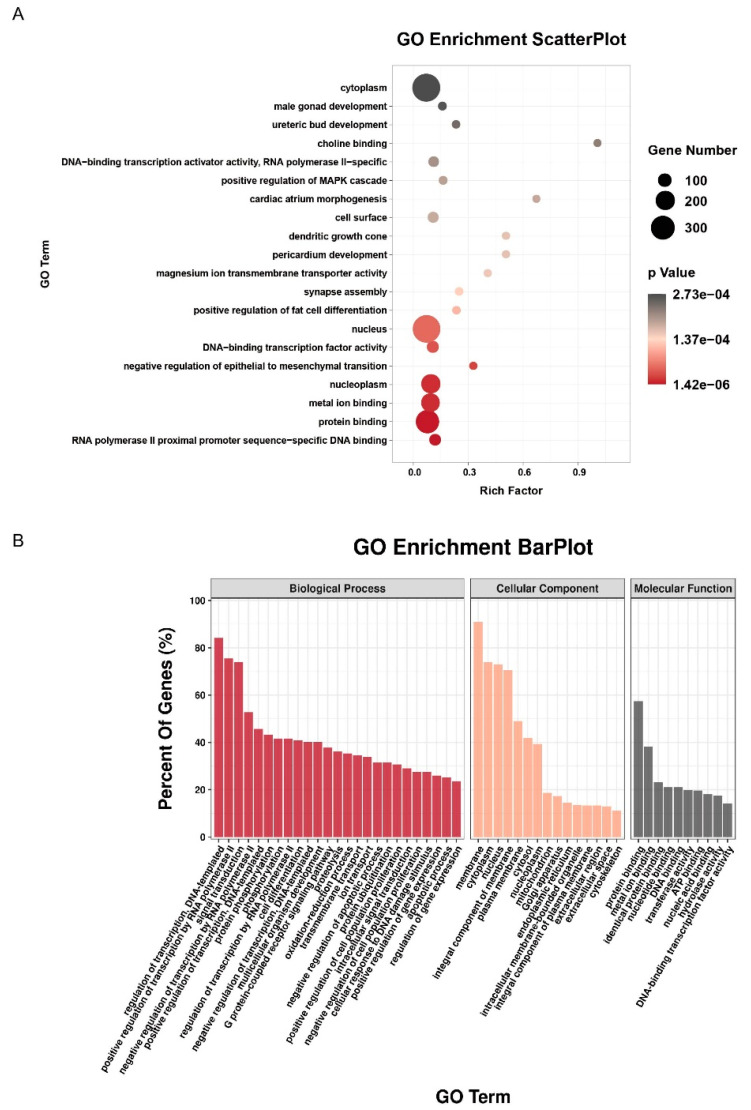
GAS5/miR-27a-3p axis regulated electrical remodeling through HOXa10. (**A**,**B**) GO enrichment analysis of the 5881 genes in the panel. (**C**) The expression of HOXa10 decreased in NRCMs after RP for 6 h, and transfection of the miR-27a-3p mimics induced the further downregulation of HOXa10. (**D**) Western blot analysis showing that si-HOXa10#2 had the best knockdown efficiency of HOXa10. (**E**) Predicted region of HOXa10 and miR-27a-3p binding sites, as yielded by luciferase activity in HEK-293T cells co-transfected with HOXa10 wild-type or mutant sequence and miR-27a-3p mimics or NC mimics. The expression of Nav1.5 and Kv4.2 in NRCMs transfected with empty vector, pcDNA-GAS5, si-HOXa10, or co-transfected with pcDNA–GAS5 and si-HOXa10 under RP, as demonstrated by RT–qPCR (**F**), western blot (**G**), and immunofluorescence assay (**H**). * *p* < 0.05, ** *p* < 0.01, *** *p* < 0.001.

**Figure 7 ijms-24-12093-f007:**
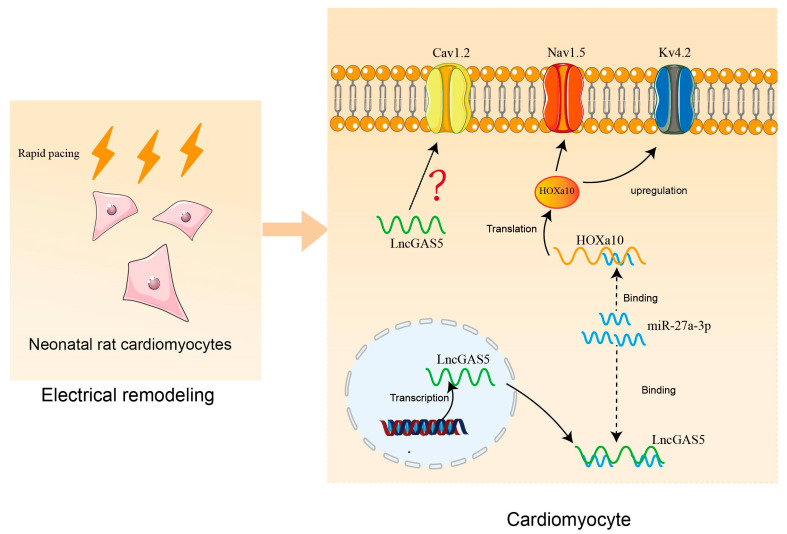
A schematic diagram showing that GAS5 mitigates RP-induced electrical remodeling via the miR-27a-3p/HOXa10 axis. In the cytoplasm of NRCMs, GAS5 targetedly binds miR-27a-3p, which decreases the expression of HOXa10 by targeting its mRNA, thereby mitigating the RP-induced downregulation of Nav1.5 and Kv4.2. RP, rapid pacing.

**Table 1 ijms-24-12093-t001:** Primer sequences.

Gene	Primer	Sequence(5′ to 3′)	
GAS5	Forward	CTGGTGGAATCTCACAGGCAG	
	Reverse	TGGCTTCCCATTCTTGTACATGG	
Cav1.2	Forward	CACAGAAGTGCAAGACACGG	
	Reverse	CCCCGCACACAATGAAACAG	
Kv4.2	Forward	GTCACCATGACAACACTGGGGTAT	
	Reverse	GATCACAGGCACGGGTAGC	
Nav1.5	Forward	GTGTCAACGGAGGTGCCAGAAC	
	Reverse	GCGTGTATGAGTGGAGTGCTTAGG	
HOXa10	Forward	AGAAGGACTCCCTGGGCAATTC	
	Reverse	CGTGTAAGGGCAGCGTTTCTTC	
GAPDH	Forward	GACATGCCGCCTGGAGAAAC	
	Reverse	ACGCTTCACGAATTTGCGTGTC	
U6	Forward	CTCGCTTCGGCAGCACATATACT	
	Reverse	ACGCTTCACGAATTTGCGTGTC	

## Data Availability

All data supporting this study are contained within the article.

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
