# Peer review of "LncRNA GAS5 Attenuates Cardiac Electrical Remodeling Induced by Rapid Pacing via the miR-27a-3p/HOXa10 Pathway"

_ijms, 2023, doi:10.3390/ijms241512093_

Round 1

Reviewer 1 Report

The authors investigated the regulatory effect of lncRNA GAS5 (GAS5) on electrical remodeling in neonatal rat cardiomyocytes (NRCMs) induced by rapid pacing (RP) at 3.0 Hz for 360 minutes. RP reduced GAS5 level and downregulated Nav1.5, Kv4.2, and Cav1.2 levels. Further study demonstrated that either miR- 27a-3p overexpression or knockdown of HOXa10 further downregulated Nav1.5, Kv4.2, and Cav1.2 expressions. GAS5 overexpression antagonized such effect in Nav1.5 and Kv4.2, but not in Cav1.2. These results indicated that in RP-treated NRCMs, GAS5 could restore Nav1.5 and Kv4.2 21expressions via miR-27a-3p/HOXa10 pathway. Based on these findings the authors believe that GAS5/miR-27a-3p/HOXa10 pathway could be a promising therapeutic target for atrial fibrillation.

Comments/Queries

1. The extrapolation of data gleaned from small animal models of AF should be done with great skepticism, and in most cases, translation requires further study in larger mammals. In this regard, what makes the authors confident that data obtained from neonatal rat cardiomyocytes subjected to rapid pacing can be extrapolated to human atrial fibrillation?

2. It is likely that electrical remodeling during AF is a compensatory mechanism of the left atrial (LA) to adapt to physiological functions and inhibit further AF. Further, LA, electrical remodeling induced by tachycardia and heart failure (HF) differs in electrophysiological properties, mainly due to differences in ion channel changes. High atrial rate induced remodeling comes from significant reduction of L-type Ca++ channels, and the inability of effectively treating Ca++ ions leads to Ca++  overload in the cytoplasm; the effective refractory period and the action potential duration are then shortened, and the formation of multiple wavelets and reentry promotes the occurrence of AF. However, the L-type Ca++ channel in HF has only a small decrease, while the decrease of K+ channel and the increase of Na+/K+ exchange can partially compensate for the decrease of L-type Ca++  channel; so, there is no significant change in the action potential duration and effective refractory period in electrical remodeling. How do the aforementioned affect the interpretation of the study findings?

Minor editing

Author Response

Response

The authors investigated the regulatory effect of lncRNA GAS5 (GAS5) on electrical remodeling in neonatal rat cardiomyocytes (NRCMs) induced by rapid pacing (RP) at 3.0 Hz for 360 minutes. RP reduced GAS5 level and downregulated Nav1.5, Kv4.2, and Cav1.2 levels. Further study demonstrated that either miR- 27a-3p overexpression or knockdown of HOXa10 further downregulated Nav1.5, Kv4.2, and Cav1.2 expressions. GAS5 overexpression antagonized such effect in Nav1.5 and Kv4.2, but not in Cav1.2. These results indicated that in RP-treated NRCMs, GAS5 could restore Nav1.5 and Kv4.2 21expressions via miR-27a-3p/HOXa10 pathway. Based on these findings the authors believe that GAS5/miR-27a-3p/HOXa10 pathway could be a promising therapeutic target for atrial fibrillation.

Comments/Queries

  1. The extrapolation of data gleaned from small animal models of AF should be done with great skepticism, and in most cases, translation requires further study in larger mammals. In this regard, what makes the authors confident that data obtained from neonatal rat cardiomyocytes subjected to rapid pacing can be extrapolated to human atrial fibrillation?

Response: Thank you for your comment. Indeed, the findings from small mammals should be discreetly extrapolated. Currently the role of lncRNA in the pathogenesis of AF remains largely unclear, and to the best of our knowledge, for the first time, our study reported the regulatory role of GAS5 in the electrical remodeling of cardiomyocytes. Our research is an exploratory study conducted in in vitro experiments by using neonatal rat cardiomyocytes to investigate the impact of GAS5 and its downstream molecules on the expression of cardiac ion channels, and yielded preliminary positive results, which suggested the possibility of extrapolation to other species. Of course, considerable studies are warranted to fill the gap between small rodents and humans, such as animal experiments using canines. Although the findings of our study suggested potential association between GAS5 and myocardial electrical remodeling, extensive studies are needed to verify its practical significance as a therapeutic target and biomarker for AF, given the fact that various targeted therapeutic approaches have been introduced but severely limited by side effects or complications.[1] In our further studies, we will continue our investigation and prudently verify the regulatory effect of GAS5 on cardiac electrical remodeling in larger AF mammal models.

This has been added in the manuscript. (Page 15, Line 259 – Page 16, Line 282)

  1. It is likely that electrical remodeling during AF is a compensatory mechanism of the left atrial (LA) to adapt to physiological functions and inhibit further AF. Further, LA, electrical remodeling induced by tachycardia and heart failure (HF) differs in electrophysiological properties, mainly due to differences in ion channel changes. High atrial rate induced remodeling comes from significant reduction of L-type Ca++ channels, and the inability of effectively treating Ca++ ions leads to Ca++  overload in the cytoplasm; the effective refractory period and the action potential duration are then shortened, and the formation of multiple wavelets and reentry promotes the occurrence of AF. However, the L-type Ca++ channel in HF has only a small decrease, while the decrease of K+ channel and the increase of Na+/K+ exchange can partially compensate for the decrease of L-type Ca++  channel; so, there is no significant change in the action potential duration and effective refractory period in electrical remodeling. How do the aforementioned affect the interpretation of the study findings?

Response: Thank you for your comment.

The mechanism of AF is complex and involves at least electrical remodeling, structural remodeling, and activation of renin angiotensin system. In patients with chronic HF, atrial structural remodeling is the primary basis for AF, causing increased conduction heterogeneity and localized conduction block, facilitating re-entry hence maintenance of AF.[2] Other studies reported that electrical coupling among cardiomyocytes can also be affected, further exhibiting the complexity of electrical remodeling in AF patients with HF. However, rapid atrial rate alone only induces electrical remodeling without causing structural remodeling, except when hemodynamic overload is present.[3] Therefore it seems reasonable that under various pathological conditions, the mechanism of AF could be different.

Our study was performed by using NRCMs, which excluded physiological variables such as mechanical stress, hormones, and autonomous nervous system. Although the confounding factors were excluded hence suitable for investigation in a single variable, such results could be considerably limited by experimental conditions. Further studies must be performed to verify the findings of our study under in vivo environment and pathological models, especially HF-induced AF model.

Our study aimed to explore the regulatory effect of GAS on electrical remodeling induced by rapid pacing, which mimicked the electrophysiological condition during AF onset, and found that (1) GAS5 is involved in myocardial electrical remodeling induced by rapid pacing; (2) a GAS5/miR-27a-3p/HOXa10 signaling pathway mediates the regulatory effect of GAS5 on cardiac ion channels, as have been summarized in Conclusion section. And based on the results of our in vitro study, we found that GAS5 could antagonize rapid pacing-induced electrical remodeling and may be a possible therapeutic target for AF. Nevertheless, given the complexity of AF pathogenesis and limitation of our study as mentioned above, we still could not draw the conclusion if GAS5 exerts a positive or negative effect under all circumstances.

Our study observed a positive association between GAS5 and AF, possibly via the miR-27a-3p/ HOXa10 pathway. Yet its positive/negative regulatory effect on electrical remodeling and consequently on the pathogenesis of AF still warrants considerable researches, especially considering that electrical remodeling during AF is a compensatory mechanism of the left atrial (LA) to adapt to physiological functions and inhibit further AF. Therefore we prudently report the association between GAS5 and rapid pacing-induced electrical remodeling, and merely propose the possibility of its potential application. Based on the current findings, we will verify the impact of GAS5 on the pathogenesis of AF in our further studies.

We revised the manuscript on English style and grammar as follows, which has been highlighted in red in the manuscript.

Page 1, Line 9, 12, 15, 16, 22, 24, 29, 30;

Page 5, Line 97, 104-106, 109;

Page 7, Line 124, 130,

Page 9, Line 163;

Page 11, Line 174;

Page 14, Line 196. 207;

Page 15, 233, 240, 243, 253;

Page 18, Line 366, 373, 382, 391, 399

  1. Babapoor-Farrokhran, S.; Gill, D.; Rasekhi, R.T. The role of long noncoding RNAs in atrial fibrillation. Heart rhythm 2020, 17, 1043-1049, doi:10.1016/j.hrthm.2020.01.015.
  2. Qiu, D.; Peng, L.; Ghista, D.N.; Wong, K.K.L. Left Atrial Remodeling Mechanisms Associated with Atrial Fibrillation. Cardiovascular engineering and technology 2021, 12, 361-372, doi:10.1007/s13239-021-00527-w.
  3. Schoonderwoerd, B.A.; Van Gelder, I.C.; Van Veldhuisen, D.J.; Van den Berg, M.P.; Crijns, H.J. Electrical and structural remodeling: role in the genesis and maintenance of atrial fibrillation. Progress in cardiovascular diseases 2005, 48, 153-168, doi:10.1016/j.pcad.2005.06.014.

Reviewer 2 Report

This article by Xi et al. aims to investigate the regulatory effect of lncRNAs GAS5 on electrical remodeling in neonatal rat cardiomyocytes induced by rapid pacing. In conclusion, the authors have deduced a GAS5/miR-27a-3p/HOXa10 signaling pathway to mediate the regulatory effect of GAS5 on cardiac ion channels. As they claimed the results of this study unveiled the new functions of GAS5 providing a theoretical basis for a new possible therapeutic target for atrial fibrillation. This is an interesting piece of work carried out systematically and well-presented. However, the following points need to be addressed before this paper could be accepted for publication:

1.    As this article involves the usage of many abbreviations, all the abbreviations should be collected in a separate list along with their full form for easy reference.

2.    One more paragraph should be added in the introduction to update with the current knowledge on this topic from the reported literature.

3.    In Figure 3A, the labels alongside top-to-bottom are too small to aid readability.

4.    A schematic diagram depicting the flow of systematic experimental & theoretical approaches adopted in this study should be included as Figure 1.

5.    A reference citation each should be provided for sections 4.5 to 4.10.

6.    The purchase details of SPSS software in section 4.11 should be included.

7.    It will be informative if the authors could add a few more lines in the conclusion section detailing the limitations/research gaps and future perspectives.

Minor editing of English language required

Author Response

Response

This article by Xi et al. aims to investigate the regulatory effect of lncRNAs GAS5 on electrical remodeling in neonatal rat cardiomyocytes induced by rapid pacing. In conclusion, the authors have deduced a GAS5/miR-27a-3p/HOXa10 signaling pathway to mediate the regulatory effect of GAS5 on cardiac ion channels. As they claimed the results of this study unveiled the new functions of GAS5 providing a theoretical basis for a new possible therapeutic target for atrial fibrillation. This is an interesting piece of work carried out systematically and well-presented. However, the following points need to be addressed before this paper could be accepted for publication:

  1. As this article involves the usage of many abbreviations, all the abbreviations should be collected in a separate list along with their full form for easy reference.

Response: Thank you for your comment. The abbreviation list has been added in the manuscript.(Page 20)

  1. One more paragraph should be added in the introduction to update with the current knowledge on this topic from the reported literature.

Response: Thank you for your comment. We thoroughly reviewed articles related to our topic and made certain supplement and modification in the Introduction section. (Page 1, Line 29-31; Page1, Line 33-49; Pabge 2, Line 46-55; Page 2, Line 57-60)

  1. In Figure 3A, the labels alongside top-to-bottom are too small to aid readability.

Response: Thank you for your comment. We adjusted the size of Figure 3A to show the labels clearly. (Page 8, Line 145)

  1. A schematic diagram depicting the flow of systematic experimental & theoretical approaches adopted in this study should be included as Figure 1.

Response: Thank you for your comment. We plotted a schematic diagram including the experimental and theoretical approaches along with the figure legend. (Figure 1 in Page 2, Line 65-68) And the numbers of other figures have been revised subsequently. (Page 3, Line 75, 82, 83-85; Page 5, Line 89, 102-114; Page 7, Line 119, 132-136, 142; Page 9, Line 146, 158, 159; Page 10, Line 166; Page 11, Line 176-188; Page 14, Line 195; Page 16, Line 206)

  1. A reference citation each should be provided for sections 4.5 to 4.10.

Response: Thank you for your comment. The references have been supplemented in Methods section 4.5 to 4.10. (Page 17, Line 329; Page 18, Line 336, 342, 356, 358, 372)

  1. The purchase details of SPSS software in section 4.11 should be included.

Response: Thank you for your comment. The purchase detail of SPSS has been added in the manuscript. (Page 19, Line 401)

  1. It will be informative if the authors could add a few more lines in the conclusion section detailing the limitations/research gaps and future perspectives.

Response: Thank you for your comment. There are still some limitations to the present study such as the in vitro study design and lack of data from large mammals. Therefore extrapolation of our findings should be prudently carried out, and further studies are warranted to verify our findings in larger mammals and explore its clinical prospect. these have been added in the Conclusion section. (Page 19, Line 409-412)

We revised the manuscript on English style and grammar as follows, which has been highlighted in red in the manuscript.

Page 1, Line 9, 12, 15, 16, 22, 24, 29, 30;

Page 5, Line 97, 104-106, 109;

Page 7, Line 124, 130,

Page 9, Line 163;

Page 11, Line 174;

Page 14, Line 196. 207;

Page 15, 233, 240, 243, 253;

Page 18, Line 366, 373, 382, 391, 399

Reviewer 3 Report

- The authors hypothesized that GAS5 regulated electrical remodeling induced by rapid pacing (RP) via miR-27a-3p/ HOXa10 pathway and used neonatal rat cardiac myocytes (NRCMs) to verify RP-induced cardiac electrical remodeling and explore the underlying mechanism.

- The study is novel and fills a gap in the current literature.

- The introduction section is too short and needs to be expanded to better explain the aim of the study. More pertinent references should be cited.

- Lines 50-51 of the introduction is related to results and should be deleted from this section.

- Results are relevant well detailed, there are good figures.

- Discussion section is too short and needs to be expanded with a more critical view by the authors.

- Methods section is clear and robust.

English should be revised for grammar and style

Author Response

Response

- The authors hypothesized that GAS5 regulated electrical remodeling induced by rapid pacing (RP) via miR-27a-3p/ HOXa10 pathway and used neonatal rat cardiac myocytes (NRCMs) to verify RP-induced cardiac electrical remodeling and explore the underlying mechanism.

- The study is novel and fills a gap in the current literature.

Response: Thank you for your comment. Our study observed a positive association between GAS5 and rapid pacing-induced electrical remodeling, possibly via the miR-27a-3p/HOXa10 pathway, which is a preliminary finding. We will continue our investigation in our further study for verification.

- The introduction section is too short and needs to be expanded to better explain the aim of the study. More pertinent references should be cited.

Response: Thank you for your comment. We expanded and revised the Introduction section to better explain the background and the aim of the study. (Page 1, Line 29-31; Page1, Line 33-49; Page 2, Line 46-55; Page 2, Line 57-60)

- Lines 50-51 of the introduction is related to results and should be deleted from this section.

Response: Thank you for your comment. The sentence mentioned above has been removed from Introduction section.

- Results are relevant well detailed, there are good figures.

Response: Thank you for your comment. It is our honor to obtain your recognition.

- Discussion section is too short and needs to be expanded with a more critical view by the authors.

Response: Thank you for your comment. We reviewed our results and expanded the discussion from a more critical view by prudently proposing the potential practical significance and thoroughly summarizing the limitation of the study.

Atrial rapid pacing could increase the susceptibility to AF via electrical remodeling,[1,2] such as abnormal calcium handling, which increases ectopic activity and shortens atrial APD and ERP,[3,4] and reduced gap junctions, which promoted atrial conduction heterogeneity.[5]

Currently, the role of lncRNA in the pathogenesis of AF remains largely unclear, and to the best of our knowledge, for the first time, our study reported the regulatory role of GAS5 in the electrical remodeling of cardiomyocytes. Our research is an exploratory study conducted in in vitro experiments by using neonatal rat cardiomyocytes to investigate the impact of GAS5 and its downstream molecules on the expression of cardiac ion channels and yielded preliminary positive results, which suggested the possibility of extrapolation to other species. However, there are some limitations that must be taken into consideration for interpretation of the results. (1) Considerable studies are warranted to fill the gap between small rodents and humans, such as animal experiments using canines. (2) Although the findings of our study suggested a potential association between GAS5 and myocardial electrical remodeling, extensive studies are needed to verify its practical significance as a therapeutic target and biomarker for AF, given the fact that various targeted therapeutic approaches have been introduced but severely limited by side effects or complications.[6] (3) The mechanism of AF is complex and involves at least electrical remodeling, structural remodeling, and autonomous nerve system.[7] And when concomitant diseases such as heart failure is present, the pathogenesis of AF tends to involve multiple mechanisms and such diversity could be enhanced during the progression of AF. Our study observed a positive association between GAS5 and AF, possibly via the miR-27a-3p/ HOXa10 pathway. Yet its regulatory effect on electrical remodeling and consequently on the pathogenesis of AF still warrants considerable researches under different concomitant diseases and at different stages of AF (such as paroxysmal and chronic AF). In our further studies, we will continue our investigation and prudently verify the regulatory effect of GAS5 on cardiac electrical remodeling in larger AF mammal models and on the pathogenesis of AF.

Hence our study suggested the possibility of a potential therapeutic target for AF, which warrants further studies for verification.

These have been supplemented in Discussion section. (Page 14, Line 215-217; Page 15, Line 221-224; Page 15, Line 259 – Page 16, Line 282)

- Methods section is clear and robust.

Response: Thank you for your comment. We really appreciate your recognition.

We revised the manuscript on English style and grammar as follows, which has been highlighted in red in the manuscript.

Page 1, Line 9, 12, 15, 16, 22, 24, 29, 30;

Page 5, Line 97, 104-106, 109;

Page 7, Line 124, 130,

Page 9, Line 163;

Page 11, Line 174;

Page 14, Line 196. 207;

Page 15, 233, 240, 243, 253;

Page 18, Line 366, 373, 382, 391, 399

  1. Wijffels, M.C.; Kirchhof, C.J.; Dorland, R.; Allessie, M.A. Atrial fibrillation begets atrial fibrillation. A study in awake chronically instrumented goats. Circulation 1995, 92, 1954-1968, doi:10.1161/01.cir.92.7.1954.
  2. Yoo, S.; Pfenniger, A.; Hoffman, J.; Zhang, W.; Ng, J.; Burrell, A.; Johnson, D.A.; Gussak, G.; Waugh, T.; Bull, S.; et al. Attenuation of Oxidative Injury With Targeted Expression of NADPH Oxidase 2 Short Hairpin RNA Prevents Onset and Maintenance of Electrical Remodeling in the Canine Atrium: A Novel Gene Therapy Approach to Atrial Fibrillation. Circulation 2020, 142, 1261-1278, doi:10.1161/circulationaha.119.044127.
  3. Herraiz-Martínez, A.; Tarifa, C.; Jiménez-Sábado, V.; Llach, A.; Godoy-Marín, H.; Colino-Lage, H.; Nolla-Colomer, C.; Casabella-Ramon, S.; Izquierdo-Castro, P.; Benítez, I.; et al. Influence of sex on intracellular calcium homoeostasis in patients with atrial fibrillation. Cardiovasc Res 2022, 118, 1033-1045, doi:10.1093/cvr/cvab127.
  4. Qi, X.Y.; Vahdati Hassani, F.; Hoffmann, D.; Xiao, J.; Xiong, F.; Villeneuve, L.R.; Ljubojevic-Holzer, S.; Kamler, M.; Abu-Taha, I.; Heijman, J.; et al. Inositol Trisphosphate Receptors and Nuclear Calcium in Atrial Fibrillation. Circ Res 2021, 128, 619-635, doi:10.1161/circresaha.120.317768.
  5. Santa Cruz, A.; MeÅŸe, G.; Valiuniene, L.; Brink, P.R.; White, T.W.; Valiunas, V. Altered conductance and permeability of Cx40 mutations associated with atrial fibrillation. The Journal of general physiology 2015, 146, 387-398, doi:10.1085/jgp.201511475.
  6. Babapoor-Farrokhran, S.; Gill, D.; Rasekhi, R.T. The role of long noncoding RNAs in atrial fibrillation. Heart rhythm 2020, 17, 1043-1049, doi:10.1016/j.hrthm.2020.01.015.
  7. Nattel, S.; Harada, M. Atrial remodeling and atrial fibrillation: recent advances and translational perspectives. J Am Coll Cardiol 2014, 63, 2335-2345, doi:10.1016/j.jacc.2014.02.555.

Round 2

Reviewer 1 Report

No further comments 

No further comments 

Reviewer 3 Report

Amended manuscript is acceptable.